# Development of Fast Sampling and High Recovery Extraction Method for Stable Isotope Measurement of Gaseous Mercury

**Satoshi Irei**

Department of Environment and Public Health, National Institute for Minamata Disease, 4058-18 Hama, Minamata, Kumamoto 867-0008, Japan; irei@nimd.go.jp; Tel.: +81-966-63-3111 (ext. 760)

**Abstract:** A method involving fast large-volume sampling and bag extraction of total gaseous mercury (TGM) using a 5 mL acid solution was developed for stable mercury isotope ratio measurements. A big gold-coated sand trap (BAuT)—a 45 (i.d.) × 300 mm (length) quartz tube with 131 times more trapping material than a conventional gold trap—was used for the collection of a large amount of TGM. The collected TGM was extracted using 5 mL inversed aqua regia in a 2 L Tedlar bag followed by isotope measurements using a cold vapor generator coupled with a multicollector inductively coupled plasma mass spectrometer. Sampling tests demonstrated that the collection efficiency of the BAuT was 99.9% or higher during the 1–24 h sampling period under the flow rate of 20–100 L min$^{-1}$. Recovery tests of 24 h bag extraction using 100 ng NIST SRM 8610 exhibited nearly 100% recovery yields. The five measured stable mercury isotope ratios agreed with reference values within 2σ intervals. The overall methodology tested during the pilot field and laboratory studies demonstrated its successful application in analysis, promising highly precise stable mercury isotopic data with a time resolution of less than 24 h.

**Keywords:** total gaseous mercury (TGM); stable mercury isotopes; multicollector inductively coupled plasma mass spectrometer (MC-ICP-MS); sampling technique; acid extraction for gaseous elemental mercury; Minamata Convention on Mercury

## 1. Introduction

Mercury is a notorious metal pollutant that can trigger serious adverse health effects [1]. It has a unique nature that it evaporates under room temperature and pressure, resulting in global spread through the atmosphere and ubiquity in the natural environment. The international regulation on the use and trade of manmade mercury, the Minamata Convention on Mercury, was enforced by the United Nations to lower the level of global mercury pollution; however, its effectiveness is questionable [2] because of the possibly significant natural sources such as volcanoes [3], the ocean surface [4], forests [5], and permafrost [6]. Recent stable mercury isotope analysis at the natural abundance has the potential to untangle the complex source apportionment that concentration measurements alone cannot solve.

There are seven stable isotopes of mercury, namely $^{196}$Hg, $^{198}$Hg, $^{199}$Hg, $^{200}$Hg, $^{201}$Hg, $^{202}$Hg, and $^{204}$Hg with typical abundances of 0.16%, 10.0%, 16.9%, 23.1%, 13.2%, 29.7%, and 6.8%, respectively [7]. These isotopic compositions can be simultaneously measured by injecting a solution containing Hg$^{2+}$ and a tin chloride solution into a cold vapor generator (CV), followed by a multicollector inductively coupled plasma mass spectrometer (MC-ICP-MS). To date, several methods to determine the stable isotopic compositions of atmospheric total gaseous mercury (TGM), which is a combination of gaseous elemental mercury (GEM) and gaseous oxidized mercury (GOM), have been reported [5,8–12]; however, these methods require long sampling durations, from a few days to weeks, thus making

source apportionment difficult. Additionally, the efficient conversion of 100 ng or more of GEM into $Hg^{2+}$ in a small amount of solution is also technically challenging due to the slow oxidation reaction. A dynamic oxidation method (i.e., slowly bubbling GEM in an acid solution) has been used for this conversion [5,8,13–15], but the reproduction of this method is difficult and strongly depends on the technical skills and knowledge of analysts. Herein, we report novel and simple offline techniques of fast large-volume sampling and high recovery extraction for stable mercury isotope measurements.

## 2. Materials and Methods

### 2.1. Sampling and Preconcentration of TGM

TGM was collected using a big gold-coated sand trap (BAuT) (Figure 1). A BAuT is a custom-made 50 (o.d.) × 45 (i.d.) × 300 mm (length) quartz tube with a 45 (o.d.) × 5 mm (thickness) fritted quartz plate (COSMOS VID, Fukuoka, Japan). Approximately 10.5 g gold-coated sand, which traps TGM at room temperature and releases TGM as GEM at high temperatures and is reused more than 50,000 times (Nippon Instruments Corporation, Osaka, Japan) was layered on the fritted plate, and approximately 0.1 g quartz wool was then stuffed on the top of the sand. The BAuT was used vertically during sampling because the trapping material settled, but was not secured onto the fritted plate. Compared to a commercially available conventional gold-coated sand trap (a 4 (i.d.) × 160 mm quartz tube with 0.08 g gold-coated silica sand, capable of capturing 70 µg of TGM; Nippon Instruments Corporation), the mouth of the BAuT is approximately 11 times wider, and the amount of trapping material is approximately 131 times greater, allowing a larger air flow with the same collection efficiency and capturing capacity of 9.2 mg of TGM. To meet the same linear velocities as those of the conventional trap under the typical flow rates of 0.5 and 1.0 L min$^{-1}$, the sampling flow rates used for the BAuT were 63.3 and 126.6 L min$^{-1}$, respectively. Techniques reported in the literature use sampling flow rates of 2–20 L min$^{-1}$ with a sampling duration of 12 h–20 days (using a carbon trap impregnated with iodine or chlorine [10,16]), 14.4 L min$^{-1}$ with 2–3 days (1.8 L min$^{-1}$ × 8 conventional gold traps [5,8]), and 16 L min$^{-1}$ with 3 days (1 L min$^{-1}$ × 16 conventional gold traps [9]). Assuming identical collection efficiencies, the BAuT sampling is 3–9-fold faster than that of the other methods. An additional advantage of the BAuT is the ease of preconditioning, its long lifetime, and the use of a single trap that makes the postsampling procedure simple.

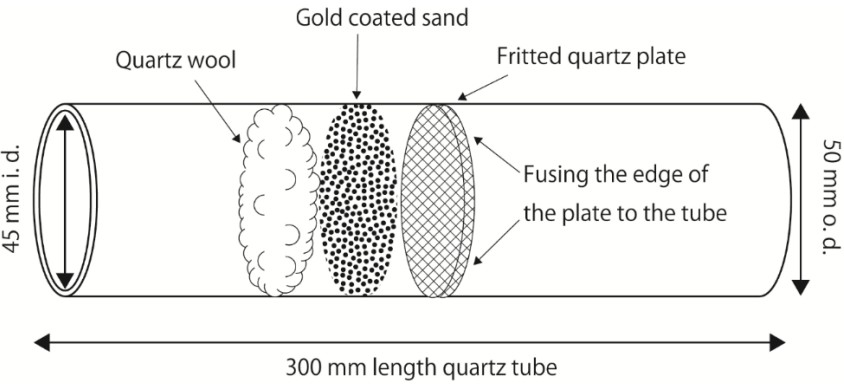

**Figure 1.** A schematic of the large gold-coated sand trap (BAuT).

TGM was sampled by the BAuT using a diaphragm pump (N860FTE, KNF, Freiburg, Germany or DOA-P501-DB, GAST MFG Corp., Benton Harbor, MI, USA) through a flow meter (RK230, KOFLOC, Kyoto, Japan). PTFE connectors for the BAuT were custom-made (COSMOS VID, Fukuoka, Japan), and half an inch or 3/8 of an inch (o.d.) PFA tubing (Tombo 9003-PFA, NICHIAS Corporation, Tokyo, Japan), PFA connectors (Swagelok, Solon, OH, USA), and/or PTFE connectors (Flowell 30 series, Flowell Corporation, Yokohama, Japan) were also used.

TGM trapped in the BAuT was preconcentrated to a conventional gold trap prior to the plastic bag extraction. The BAuT was heated to 873 K for 1 h under a 0.5 L min$^{-1}$ zero air flow, which was supplied by compressed room air (super oil free BEBICON 0.4LE-8SB, Hitachi Industrial Equipment Systems Co., Ltd., Tokyo, Japan) through a dryer stuffed with silica gel (Kanto Chemical Co., Inc., Tokyo, Japan) and a mercury trap stuffed with activated charcoal (Hokuetsu MA-HG, Ajinomoto Fine Techno Co., Inc., Yokohama, Japan). The background TGM concentration in the zero air was 0.03 pg L$^{-1}$ or less. The flow rate was controlled using a mass flow controller (SEC-E40 and PE-D20, HORIBA, Ltd., Kyoto, Japan). The BAuT was heated by a handmade heater consisting of a nickel-chrome wire (1.0 mm (o.d.), Sunko Corporation, Kyoto, Japan), quartz wool (GL Sciences, Tokyo, Japan), insulating glass tape (Komeri Corporation, Ltd., Niigata, Japan), variable transformer (RAS-10, Tokyo Rikosha Co., Ltd., Saitama, Japan), and temperature controller (TR-KN, AS ONE Corporation, Osaka, Japan). After preconcentration, the BAuT was baked at 853 K in a muffle furnace (FUL230BL, Toyo Seisakusho Kaisha, Ltd., Chiba, Japan) for several days to remove organic and mercury contaminants. Prior to sampling, the background mercury in the BAuT was preconcentrated to a conventional trap under 873 K for 1 h with a 0.5 L min$^{-1}$ zero air flow. Then, it was quantitatively analyzed by a cold vapor atomic florescent spectrometer (CV-AFS) (Nippon Instruments Corporation) which was calibrated with saturated mercury vapor at a specific temperature (MB-1, Nippon Instruments Corporation). The observed background mercury was less than 70 pg.

### 2.2. Plastic Bag Extraction of TGM

Preconcentrated TGM was extracted with a 5 mL trapping solution in a plastic bag. Plastic bags were chosen for this purpose because, compared to a hard container, they are light and easy to evacuate and fill with the gas and have a large inner surface area, which is needed to increase the contact area of the trapping solution with the sample gas. The preconcentrated TGM in the conventional trap was flushed into a 2 L Tedlar (polyvinyl fluoride) bag through the attached PTFE stopcock (AS ONE Corporation, Osaka, Japan) for 4 min under 973 K and a 0.5 L min$^{-1}$ zero air flow using another handmade heater. To enable efficient transfer, flushing was started as soon as the temperature reached 573 K. Prior to the transfer, 5 mL inversed aqua regia, a 2:1 (*v/v*) mixture of nitric and hydrochloric acids (PMA grade, Kanto Chemical Co., Inc., Tokyo, Japan) in ultrapure water (Milli-Q Direct 3, Merck KGaA, Darmstadt, Germany), was pipetted into the bag. The plastic bag was shaken for 3 min, then left for 3-24 h. The bag was shaken for 1 min three more times during the rest of the period. The bag was also weighed at the start and at the end of the extraction using an electronic balancer (ML4002T, Mettler-Toledo GmbH, Greifensee, Switzerland) to check for leaks during the extraction.

### 2.3. CV–MC–ICP–MS Measurements

The mercury isotope ratio is expressed using the following delta notation:

$$\delta^{x}\mathrm{Hg}\;(‰) = \left[ \frac{\left( \frac{^{x}\mathrm{Hg}}{^{198}\mathrm{Hg}} \right)_{sample}}{\left( \frac{^{x}\mathrm{Hg}}{^{198}\mathrm{Hg}} \right)_{reference}} - 1 \right] \times 1000 \tag{1}$$

where x and the bracketed isotope ratios with sample and reference stand for the mass of the mercury isotope and the mercury isotope ratios of the mass x to the mass 198 for the sample and reference materials, respectively. The standard reference material 3133 (or SRM 3133; NIST, Gaithersburg, MD, USA) was used for the reference material.

The following is a brief description of the stable mercury isotope measurement using a CV (HGX-200, Teledyne CETAC Technologies, Omaha, NE, USA), followed by an MC–ICP–MS (Neptune Plus, Thermo Fisher Scientific GmbH, Bremen, Germany). A sample solution and a 5% (*w/w*) SnCl$_2$ in 10% (*v/v*) hydrochloric acid solution, which instantaneously reduces Hg$^{2+}$ in a solution to GEM, were introduced into the CV by a peristaltic pump (Perimax, Spetec GmbH, Erding, Germany) under

a flow rate of 0.58 mL min$^{-1}$. Thallium standard reference material (SRM 997, NIST, Gaithersburg, MD, USA), of which the isotope ratio of $^{203}$Tl to $^{205}$Tl is known, was also introduced into the CV in order to correct artificial mass-dependent isotope fractionations as they occurred on the surfaces of the mercury and thallium at the ICP. This injection was made by introducing the dried thallium chloride aerosols generated from the hydrochloric acid/thallium (SRM 997, NIST, Gaithersburg, MD, USA) solution using an aerosol generator (Aridus II, Teledyne CETAC Technologies, Gaithersburg, NE, USA), which was operated with a 100 μL min$^{-1}$ PFA nebulizer (SP820A, Teledyne CETAC Technologies) under 383 K for the spray chamber and a desolvator, with 0.40 mL min$^{-1}$ and 2.9 L min$^{-1}$ argon flow for the nebulizer and desolvator, respectively. Argon carrier gas for the CV (known as the "counter flow" for a straight gas–liquid separator) flowed into the CV from one end at 0.56 mL min$^{-1}$ and flowed out from the other end so that all gases and aerosols inside the CV, including the sample GEM and the generated thallium aerosols, were carried into the MC-ICP-MS system. The MC-ICP-MS was tuned on a daily basis using a 10 ng g$^{-1}$ solution of SRM 3133 (NIST, Gaithersburg, MD, USA) prepared in 5% inversed aqua regia (2:1 nitric acid/hydrochloric acid) prior to the measurements. A sample was preanalyzed by the CV-MC-ICP-MS to determine the concentration. When the concentration was higher than 10 ng g$^{-1}$, the sample was diluted to 10 ng g$^{-1}$—a concentration equivalent to the reference solution. When the concentration was below 10 ng g$^{-1}$, the concentration of the reference solution was diluted to an identical concentration of the sample. The typical sensitivity of the instruments under the conditions above was 0.19 V min·g ng$^{-1}$ mL$^{-1}$ for $^{202}$Hg. This value is low compared to other reported values (e.g., 1 V min·g ng$^{-1}$ mL$^{-1}$). The variation in the sensitivity was very likely caused by the use of different instruments (i.e., CV-MC-ICP-MS).

For a sample or reference run, the operation software for the MC-ICP-MS was programmed to preinject a sample solution for 3 min for flushing, to measure the isotope ratios in 50 cycles and one block, and then to average out the measurement results. A 10 ng g$^{-1}$ standard solution of SRM 3133 in 5% inversed aqua regia was measured before and after the sample run (i.e., the bracketing method), and the average of those reference isotope ratios was used as the reference isotope ratio in the calculation of Equation (1).

To check the performance of the instrument, the dependence of the measurement accuracy on the signal intensity (or sample concentration) was evaluated using SRM 8610 (NIST, Gaithersburg, MD, USA) and SRM 3133 in different concentrations dissolved in 5% inversed aqua regia. We called this the linearity of the isotope measurements. In this evaluation, the concentrations of SRM 8610 were identical to those of SRM 3133, the reference solution. The measurement results show that the offsets of the isotope ratios from the reference values (offset = measured isotope ratio − reference isotope ratio) or the measurement accuracy for $\delta^{199}$Hg, $\delta^{200}$Hg, $\delta^{201}$Hg, $\delta^{202}$Hg, and $\delta^{204}$Hg were less than ± 0.1‰ when the concentrations were between 2.5 and 10 ng g$^{-1}$, and their uncertainties indicate that the differences are insignificant (Figure 2). Depending on the mass, the offset became slightly larger as the concentration decreased. For samples with a mercury concentration of 1.3 or 0.8 ng g$^{-1}$, the offset was as large as 0.18‰ ± 0.09‰ or −0.33‰ ± 0.12‰, respectively. The offsets were insignificant as 2σ confidence intervals of the uncertainties were considered for the 1.3 ng g$^{-1}$ solution and 3σ intervals for the 0.8 ng g$^{-1}$ solution. We conclude that the high accuracy and precision can be maintained when the sample concentrations are higher than 2.5 ng g$^{-1}$, and that the precision decreases when the concentration falls below 2.5 ng g$^{-1}$.

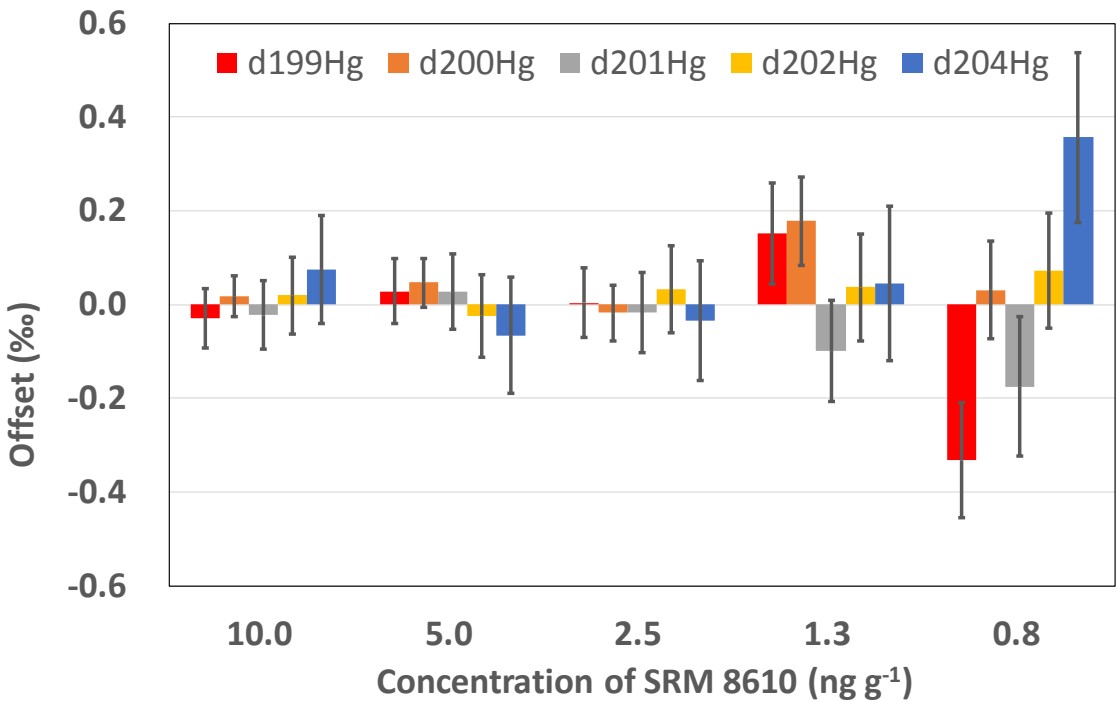

**Figure 2.** Dependence of the accuracy of stable mercury isotope measurements of SRM 8610 on the concentration. "Offset" indicates the difference between the measured and reference isotope ratios (offset = measured-reference). Error bars refer to the propagated errors estimated from the uncertainty given for SRM 8610 by NIST and the standard errors from the 50 replicate measurements.

*2.4. Methodology Test*

We performed tests for the sampling and plastic bag extraction in laboratory and field studies. The BAuT performance was evaluated by the collection efficiency: TGM was sampled using two BAuTs connected in series (double BAuT sampling), and the collection efficiency was then determined by comparing the mass of TGM in the backup BAuT with that in the primary BAuT. During this evaluation, the concentration of the preconcentrated TGM in the backup BAuT was directly measured by CV-AFS, and that in the primary BAuT was measured by CV-MC-ICP-MS. The extraction tests were performed by introducing 103 ng GEM prepared from SRM 8610 into the Tedlar bag. The dependence of the recovery yields on different inversed aqua regia concentrations (20%–40%) and the extraction time (3–24 h) was evaluated. The concentrations were chosen based on the preliminary tests, which showed that the use of 5% inversed aqua regia required 7 days to yield 90% GEM and that concentrations higher than 40% resulted in lower signal intensity (see subsection S1 in the Supplementary Materials).

The overall methodology was evaluated in source, air quality, and laboratory studies. TGM from the biomass burning was sampled on 24 March and 6 April 2019 during the Aso open field burning, also referred to as Noyaki (Figure 3). The air was drawn through a 47 mm (o.d.) Teflon-coated glass fiber filter (Pallflex Emfab, Pall Corporation, Port Washington, NY, USA) placed on an open-face filter holder (NL-O-01, NILU, Kjeller, Norway) to remove the soot. The electric power for the sampling pump was supplied from an uninterruptible power supply (SURTA1500XLJ, APC, West Kingston, RI, USA). The background concentration of TGM in the regional air was also collected on 31 March and 23 May 2019. Moreover, one outdoor and two indoor air samplings were also conducted at the institute the author affiliates. The BAuT was also tested for sampling GEM in the gas mixture used in the laboratory experiment. The experimental conditions were similar to those reported in [17,18]. Briefly, approximately 200 ng of GEM prepared from SRM 8610, oxidant precursors, and dried zero air were introduced into a 1.5 $m^3$ photochemical reactor. Then, 366 nm UV lamps (UV lamp 4, 8 W, CAMAG,



Muttenz, Switzerland) were switched on and off to initiate and stop the reactions with OH radicals, one of the major sinking pathways of atmospheric GEM. After the reaction, GEM in the gas mixture was sampled using the BAuT through a KCl-coated eight-channel annular denuder (URG-2000-30CF, URG Corporation, Chapel Hill, NC, USA), which removes GOM [19]. The sampling conditions are presented in Supplementary Table S1, together with the weather conditions. The preconcentration and extraction procedures were the same as those referred to previously.

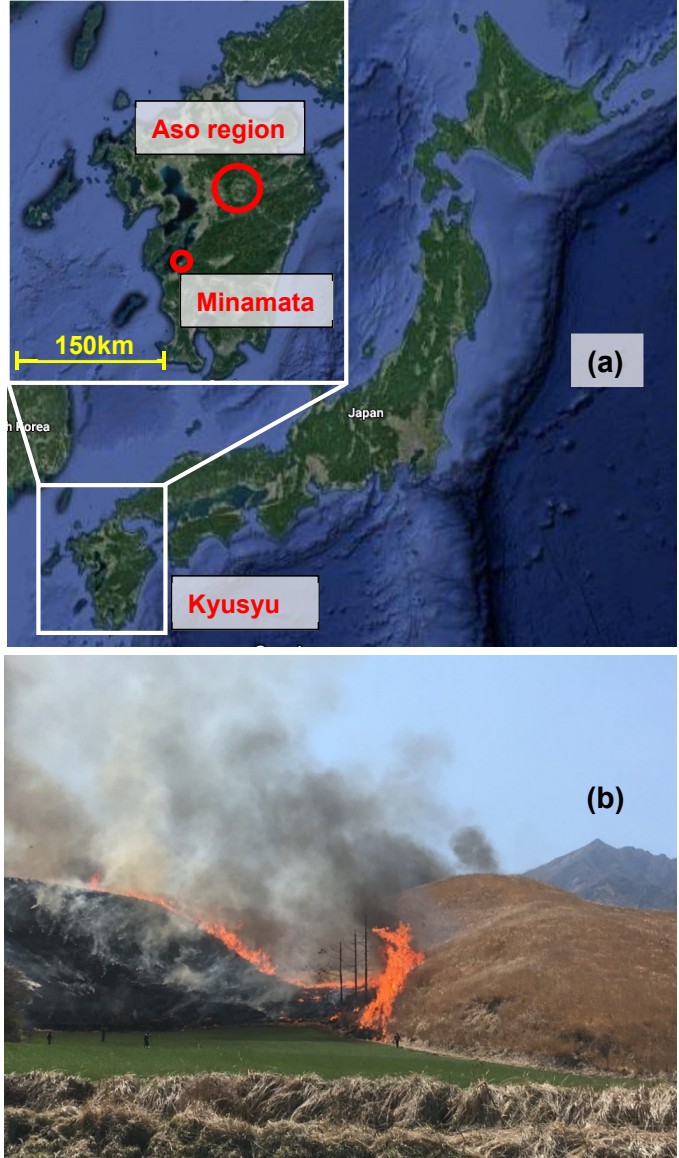

**Figure 3.** (**a**) A map of the Aso region in Kyusyu, Japan (Google Maps) and (**b**) a photo of the Aso open field burning (Noyaki) in 2019. This annual event in spring burns 161 km$^2$ of the natural grass field and stock farms to sustain healthy grass fields in this region.

## 3. Results and Discussion

### 3.1. The BAuT Collection Efficiency

The breakthrough tests performed with double BAuT showed that only 7–115 pg of TGM passed through the primary BAuT during the 1.3–24 h sampling period under a 20–100 L min$^{-1}$ flow rate (Table 1). Compared to the TGM mass trapped in the primary BAuT, the breakthrough TGM found

in the backup BAuT was negligible, corresponding to a collection efficiency of 99.9%, satisfying our demand for the fast large-volume sampling of TGM in the air.

**Table 1.** Results of the total gaseous mercury (TGM) breakthrough test.

| Sample | Sampling Duration (h) | Flow Rate (L min$^{-1}$) | Total Volume (m$^3$) | Hg in Backup BAuT (pg) | Collection Efficiency [†] (%) |
|---|---|---|---|---|---|
| Laboratory air | 6 | 100 | 36 | 7 | 99.99 |
| Clean room air | 24 | 100 | 144 | 115 | 99.86 |
| Reactor air | 1.3 | 20 | 1.5 | 74.8 | 99.93 |

[†] Collection efficiency was determined based on the trapped TGM in the backup and primary BAuTs.

### 3.2. Plastic Bag Extraction

The average ± standard deviation of the blank plastic bag extracts (*n* = 12) was 55 ± 61 pg g$^{-1}$, and all of the recovery yields determined hereafter were blank-corrected. The extraction tests exhibited no significant loss of the trapping solution (see subsection S2 in the Supplementary Materials) and no significant dependency of the recovery yields on the tested inversed aqua regia concentrations. However, it is worth noting that the preliminary tests showed that the use of 5% inversed aqua regia required 7 days to achieve 90% recovery yields. Moreover, there was the slight dependency of the recovery yields on the extraction time (Table 2): the recovery yields were 100% when the extraction time was 24 h, but slightly decreased and varied when the extraction time was shorter than 24 h. The stable mercury isotope measurements of the extracts showed that, overall, the differences from the reference values were in the range of −0.03‰ to 0.02‰ for $\delta^{199}$Hg, −0.04‰ to 0.11‰ for $\delta^{200}$Hg, −0.06‰ to 0.18‰ for $\delta^{201}$Hg, −0.07‰ to 0.23‰ for $\delta^{202}$Hg, and −0.12‰ to 0.33‰ for $\delta^{204}$Hg (Table 2). Ninety percent of the measured isotope ratios agreed with the reference values within 1$\sigma$ confidence interval and 96% within 2$\sigma$ confidence intervals. However, the precision of the samples with a 3 h extraction time was slightly poorer than that of the other extraction times. Overall, we conclude that the extraction time of 24 h with an inversed aqua regia concentration between 20% and 40% gave the best results and is recommended for plastic bag extraction. Comparison with the other extraction methods demonstrates that our method gave comparable results with those by the reported methods (Table 3). The advantages here are: the use of inversed aqua regia does not require a quenching reagent for oxidants, which is required when other strong oxidizing solutions, such as KMnO4 and BrCl, are used and slowly reduces Hg$^{2+}$, causing unintentional loss of Hg$^{2+}$ and: a number of sample extracts can be prepared in a day when TGM samples collected by the BAuTs were already preconcentrated.

**Table 2.** Results (average ± standard error) of the plastic bag extraction tests using 103 ng gaseous elemental mercury prepared from SRM 8610.

| Sample | n | Spiked Hg (ng) | Extraction Duration (h) | Recovery Yield (%) | $\delta^{199}$Hg [§] (‰) | $\delta^{200}$Hg [§] (‰) | $\delta^{201}$Hg [§] (‰) | $\delta^{202}$Hg [§] (‰) | $\delta^{204}$Hg [§] (‰) |
|---|---|---|---|---|---|---|---|---|---|
| Referenc $\delta^{x}$Hg for SRM 8610 [†] | | N.A. [‡] | N.A. [‡] | N.A. [‡] | −0.17 ± 0.01 | −0.27 ± 0.01 | −0.46 ± 0.02 | −0.56 ± 0.03 | −0.82 ± 0.07 |
| 40% inversed aqua regia | 6 | 103 ± 2 | 24 | 102 ± 5 | −0.19 ± 0.05 | −0.30 ± 0.06 | −0.51 ± 0.06 | −0.62 ± 0.09 | −0.91 ± 0.07 |
| 30% inversed aqua regia | 4 | 103 ± 2 | 24 | 100 ± 2 | −0.20 ± 0.01 | −0.30 ± 0.03 | −0.52 ± 0.02 | −0.63 ± 0.04 | −0.92 ± 0.05 |
| 20% inversed aqua regia | 3 | 103 ± 2 | 24 | 103 ± 3 | −0.10 ± 0.02 | −0.18 ± 0.03 | −0.40 ± 0.03 | −0.45 ± 0.06 | −0.67 ± 0.06 |
| 40% inversed aqua regia | 3 | 103 ± 2 | 12 | 97 ± 2 | −0.19 ± 0.05 | −0.29 ± 0.05 | −0.48 ± 0.09 | −0.59 ± 0.08 | −0.94 ± 0.15 |
| 30% inversed aqua regia | 3 | 103 ± 2 | 12 | 99.1 ± 0.5 | −0.20 ± 0.01 | −0.31 ± 0.03 | −0.52 ± 0.03 | −0.62 ± 0.05 | −0.92 ± 0.06 |
| 40% inversed aqua regia | 3 | 103 ± 2 | 6 | 97 ± 2 | −0.11 ± 0.03 | −0.18 ± 0.02 | −0.36 ± 0.06 | −0.44 ± 0.03 | −0.67 ± 0.03 |
| 30% inversed aqua regia | 3 | 103 ± 2 | 6 | 99 ± 2 | −0.19 ± 0.05 | −0.27 ± 0.05 | −0.44 ± 0.08 | −0.55 ± 0.06 | −0.87 ± 0.08 |

**Table 2.** *Cont.*

| Sample | n | Spiked Hg (ng) | Extraction Duration (h) | Recovery Yield (%) | $\delta^{199}$Hg [§] (‰) | $\delta^{200}$Hg [§] (‰) | $\delta^{201}$Hg [§] (‰) | $\delta^{202}$Hg [§] (‰) | $\delta^{204}$Hg [§] (‰) |
|---|---|---|---|---|---|---|---|---|---|
| 20% inversed aqua regia | 2 | 103 ± 2 | 6 | 92 ± 6 | −0.15 ± 0.01 | −0.19 ± 0.05 | −0.28 ± 0.09 | −0.33 ± 0.14 | −0.49 ± 0.34 |
| 40% inversed aqua regia | 3 | 103 ± 2 | 3 | 87 ± 11 | −0.15 ± 0.01 | −0.21 ± 0.03 | −0.38 ± 0.04 | −0.46 ± 0.08 | −0.69 ± 0.21 |
| 30% inversed aqua regia | 2 | 103 ± 2 | 3 | 96 ± 5 | −0.15 ± 0.05 | −0.24 ± 0.09 | −0.44 ± 0.17 | −0.50 ± 0.20 | −0.74 ± 0.30 |

[†] Recommended isotope ratios according to NIST. [‡] N.A.: not applicable. [§] The measured $\delta^x$Hg values shown are average ± standard errors form the 50 replicate measurements.

**Table 3.** Comparison of acid extraction methods for gaseous elemental mercury.

| | This Method | Yamakawa et al. [14] | Sun et al. [13] | Demers et al. [5] | Janssen et al. [15] |
|---|---|---|---|---|---|
| Tested Hg quantity (ng) | 103 | unknown | 0.5–2.7 | 30–80 | 4.7–200 |
| Type of acid used for the extraction | 40% inversed aqua regia | 1% $KMnO_4$ in 5% $H_2SO_4$ | inversed aqua regia | 1% $KMnO_4$ | 0.03–1% $KMnO_4$ or 40% $HNO_3$:BrCl |
| Extraction time (h) | 24 | 2 | unknown | unknown | 0.7 |
| Extraction method | plastic bag | purging under the programmed heating | purging under the programmed heating | purging under the programmed heating | purging under the programmed heating |
| Recovery yield (%) | 100–103 | 99.1–99.6 | 75–102 | 89–94 | 96–101 |

## 3.3. Methodology Tests

Analysis of the ambient samples collected in the Aso region showed that the atmospheric TGM concentrations during Noyaki on 24 March and 6 April were 3.7 and 1.4 ng m$^{-3}$, respectively, while those in the background air were 0.7 and 0.8 ng m$^{-3}$, respectively (Table 4), indicating that the Noyaki events increased the TGM concentrations. The TGM concentrations in the background air were slightly lower than the expected—approximately 1 ng m$^{-3}$—which is the minimum GEM concentration observed at the Japanese background sites [20]. The filtering of the air through the PTFE filter kept the trapping material clean and avoided its deterioration; however, airborne particulate matter trapped on the PTFE filter may have adsorbed TGM and lowered the TGM concentration. The obtained extract concentrations of the Noyaki and background air samples using the plastic bag extraction method were in the range of 3.1–1.0 ng g$^{-1}$, and the results of the isotope ratio measurements for these analyses are shown in Table 3. It should be noted that according to the linearity of the isotope measurements discussed in Section 2.3., the $\delta^x$Hg for the 6 April sample may have up to a ± 0.4‰ bias. The samples collected during the Noyaki events were mixtures of TGM from Noyaki and background air; therefore, we subtracted the averaged background concentrations and isotopic compositions to calculate the true $\delta^x$Hg of TGM emitted during the Noyaki using the mass balance for the isotope ratios [21]. Except for $\delta^{204}$Hg of the 6 April sample, the background-corrected $\delta^x$Hg values were of light isotopic compositions (Table 5), which may refer to the isotope fractionation associated with the burning during Noyaki. To further investigate the observed $\delta^x$Hg values, we compared them to the predicted $\delta^x$Hg values based on the theory of mass-dependent fractionation using the $\delta^{202}$Hg value and the scaling factor for the mass x, $^x\beta$. The deviation from the predicted value—$\Delta^x$Hg—is an indicator for evaluating the mass-dependent and independent isotope fractionations (MDF and MIF, respectively) and can be approximated as follows when $\delta^x$Hg values are smaller than 10‰ [7]:

$$\Delta^x\text{Hg} \approx \delta^x\text{Hg} - \delta^{202}\text{Hg} \times {}^x\beta \tag{2}$$

**Table 4.** Results of the overall methodology tests.

| Sample | Flow Rate (L min$^{-1}$) | Sampling Duration (h) | Air Volume (m$^3$) | Conc. of Extract [†] (ng g$^{-1}$) | Conc. in the Air [‡] (ng m$^{-3}$) | $\delta^{199}$Hg (‰) | $\delta^{200}$Hg (‰) | $\delta^{201}$Hg (‰) | $\delta^{202}$Hg (‰) | $\delta^{204}$Hg (‰) |
|---|---|---|---|---|---|---|---|---|---|---|
| Noyaki, 24 March | 75 | 0.9 | 4.2 | 3.1 | 3.7 | −0.58 | −0.58 | −1.01 | −0.93 | −1.40 |
| Noyaki, 6 April | 78 | 0.8 | 3.5 | 1.0 | 1.4 | −1.14 | −0.69 | −1.25 | −1.39 | 1.19 |
| Background air, 31 March | 54 | 2.1 | 6.7 | 1.1 | 0.8 | −0.05 | 0.44 | 0.35 | 1.06 | 1.30 |
| Background air, 23 May | 80 | 4.2 | 20.1 | 3.0 | 0.7 | −0.02 | 0.13 | −0.06 | 0.16 | 0.49 |
| The average of the background air | N.A. [§] | N.A. [§] | N.A. [§] | N.A. [§] | 0.8 | −0.04 | 0.28 | 0.14 | 0.61 | 0.89 |
| Outdoor air | 75 | 22.3 | 100.2 | 17.6 | 0.9 | −0.12 | 0.08 | 0.01 | 0.29 | 0.48 |
| Laboratory air | 100 | 6.0 | 36.0 | 23.2 | 3.2 | −0.12 | −0.11 | −0.22 | −0.12 | −0.16 |
| Clean room air | 100 | 24.0 | 144.0 | 17.0 | 0.6 | −0.23 | −0.20 | −0.45 | −0.40 | −0.59 |

[†] The concentrations of sample extracts were determined by the preanalysis of the extracts using CV–MC–ICP–MS. [‡] TGM concentrations in the air were determined according to the extract concentrations, the volume of each extract, and the sampled air volume. [§] Not applicable.

**Table 5.** Background-corrected $\delta^x$Hg and $\Delta^x$Hg for TGM from the open field burning (Noyaki).

| Sample ID | $\delta^{199}$Hg (‰) | $\delta^{200}$Hg (‰) | $\delta^{201}$Hg (‰) | $\delta^{202}$Hg (‰) | $\delta^{204}$Hg (‰) | $\Delta^{199}$Hg (‰) | $\Delta^{200}$Hg (‰) | $\Delta^{201}$Hg (‰) | $\Delta^{204}$Hg (‰) |
|---|---|---|---|---|---|---|---|---|---|
| Noyaki, 24 March | −0.73 | −0.82 | −1.33 | −1.35 | −2.03 | −0.39 | −0.14 | −0.31 | −0.01 |
| Noyaki, 6 April | −2.53 | −0.72 | −0.84 | −1.18 | 1.10 | −2.24 | −0.13 | 0.05 | 2.87 |

The reported scaling factors for the masses of 199, 200, 201, and 204 were 0.2520, 0.5024, 0.7520, and 1.493, respectively. The calculated $\Delta^x$Hg values for the background-corrected TGM from the 24 March Noyaki showed a slightly, but significantly, odd MIF (Table 5). This may imply that such light isotopic compositions may be associated with the isotopic fractionations that occur in the evaporation process during the burning of a field, and the process may involve a small MIF. Such an MIF was not observed in the 6 April Noyaki sample, and the $\Delta^{199}$Hg and $\Delta^{204}$Hg varied widely, i.e., −2.3‰ and 2.9‰, respectively. The reason for the variation is unknown, and further research is needed.

Analysis of the TGM from the laboratory and clean room air showed the sufficient collection of TGM (116 and 85 ng, respectively) and the successful analysis of their $\delta^x$Hg values, ranging from −0.11‰ to −0.59‰ (Table 4). The collection of GEM from the outdoor air was also successful (88 ng) and sufficient for creating a sample solution for isotope measurements, showing a range from −0.12‰ to 0.48‰. The $\delta^x$Hg values in the clean room air were the lightest, and this may be attributed to the isotope fractionation that occurred at the surface of mercury absorbent used for cleaning the air. The $\Delta^x$Hg values for all of these air samples showed a consistent trend—a higher $\Delta^x$Hg value for a greater mass (Table 6).

**Table 6.** Calculated $\Delta^x$Hg values for the air and laboratory experimental samples.

| Sample | $\Delta^{199}$Hg (‰) | $\Delta^{200}$Hg (‰) | $\Delta^{201}$Hg (‰) | $\Delta^{204}$Hg (‰) |
|---|---|---|---|---|
| Outdoor air | 0.11 | 0.55 | 0.71 | 1.87 |
| Laboratory air | 0.12 | 0.36 | 0.48 | 1.23 |
| Clean room air | 0.00 | 0.27 | 0.25 | 0.80 |
| Reactor GEM | 8.96 | −1.43 | 5.44 | 10.52 |

The $\delta^x$Hg of the residual GEM after the reaction with OH radicals exhibited large isotope fractionations—from −1.63‰ to 9.96‰ at the 44% extent of Hg reaction estimated from the concentration of the extract (Table 4). A saw-tooth pattern was clearly seen in the $\delta^x$Hg profile between $^{199}$Hg and $^{202}$Hg. With the exception of $^{204}$Hg, this pattern was due to MDF and MIF on odd- and even-numbered masses, respectively: for the odd-numbered masses, the light mercury isotope—$^{198}$Hg—reacted faster than the heavy isotopes of $^{199}$Hg and $^{201}$Hg, resulting in heavy $\delta^x$Hg values in the residual GEM; meanwhile, for the even-numbered masses, the heavy isotopes of $^{200}$Hg and $^{202}$Hg reacted faster than $^{198}$Hg, resulting in light $\delta^x$Hg values. The $\delta^{204}$Hg value is an exception, as this even-numbered

mass isotope showed strong MDF. The calculated $\Delta^x$Hg for these isotopes showed this trend more clearly (Table 6). The unusual MIF of $^{200}$Hg found in the atmospheric precipitation of Peterborough, Canada [22] may be explained by this process, but the strong MDF found for the odd-numbered masses here does not explain their observations. Again, thorough evaluation is needed of these kinetic isotope effects.

## 4. Conclusions

We developed a method involving fast and robust large-volume sampling for TGM and novel high-recovery extraction for stable mercury isotope analysis. The sampling tests demonstrated 99.9% collection efficiency of TGM by the BAuT under a flow rate of 20–100 L min$^{-1}$ over a sampling duration of 1.3–24 h. The recovery tests of the plastic bag extraction method using 100 ng of GEM prepared from NIST SRM 8610 showed almost 100% of the recovery yields, which were reproducible when the extraction time was 24 h and 40% inversed aqua regia was used in the extractions. The developed techniques here are simple and promising, ideal for source and laboratory studies where fast sampling is needed. These sampling and preconcentration techniques can also be automated, making them ideal for monitoring studies of stable mercury isotopes at remote sites.

**Supplementary Materials:** The following are available online at http://www.mdpi.com/2076-3417/10/19/6691/s1.

**Funding:** This project was financially supported by the internal competitive funding of the National Institute for Minamata Disease (RS-19-17, RS-19-19, and RS-20-11).

**Acknowledgments:** The author acknowledges Nanami Yamamoto for her assistance in this project.

**Conflicts of Interest:** The authors declare no conflict of interest.

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
