# Peer review of "Development of Fast Sampling and High Recovery Extraction Method for Stable Isotope Measurement of Gaseous Mercury"

_applsci, doi:10.3390/app10196691_

Round 1

Reviewer 1 Report

The manuscript describes a method for determination of mercury isotopes in air with high temporal resolution. The method is novel in two features: the use of large volume Au trap enabling high sampling flow coupled with an extraction method using a Tedlar bag.

The method and its tests are described in sufficient detail to be replicated. The manuscript, however, suffers from awkward and at times unclear wording due to poor English. I recommend its publication after a thorough editing by a native English speaker.

Notes and recommendations:

Keywords are missing.

Recommendation: The BAuT should always be used in vertical position. In horizontal position the sand could settle down by e.g. pump vibrations or during the transport which would create channels for the sampled air without much contact wit Au surface. A plug of quartz wool is not sufficient to prevent such settling. Sampling through horizontal BAuT could be the reason for low background concentrations of 0.7 and 0.8 ng m-3, unusually low for northern hemisphere.

Reference 18 not mentioned in the text. Neither it seems to be relevant for this paper, according to its title.

The manuscript needs urgently to be edited by a native English speaker. The English is awkward and at times difficult to understand. Some examples:

Line 15: “The sampling test by .. ” could be replaced by “The collection efficiency of BAuT was 99.9% or higher during…”

Lines 38-41: “..isotopic compositions … have been reported. Their methods …”?

Line 42: “technically challenging” instead of “technical challenging”

Lines 54-57: The sentence is not correct, probably it should be two sentences.

Line 73: “… to see the concentration.”?

Line 78: “The low sensitivity here very likely attributes…”?

Line 93: “… and their uncertainties exhibit that …”?

Line 98: “if” instead of “as”

Line 140: “home-made” instead of “hand-made”. Also lines 152-153.

Line 151: “Tedlar” instead of “Tedler”

Line 174: “The overall methodology was evaluated in source, air quality, and laboratory studies”? Source studies?

Lines 176-178: “... set on …”

Line 226: “… during the Noyaki …”? What is Noyaki?

Reviewer 2 Report

This is an interesting and novel methodology looking at a (very) high flow rate technique for atmospheric monitoring of stable Hg isotopes. The author is clearly a technically gifted and creative scientist and this work is a credit to them. Overall, I would make a few suggestions before accepting this work for publication.

  1. While it is a credit to the author to achieve this work on their own, the single authorship does show-up negatively a little in the writing. The manuscript can be convoluted, the message sometimes a little unclear, and some acronyms and place holders confusing and unnecessary. Unfortunately, this is a side affect of writing a paper on your own. It would be good if the author could get some help to polish some of the writing for clarity, and even some minor restructuring from someone at a distance from the work. Some recommendations are made below.
  2. One issue that I see is with QA/QC of the BAuT sampling method. The in-line BAuTs recoveries are tested with different methods. The sampling trap with the MC-ICPMS and the breakthrough trap with CV-AFS. Similarly, there was no concurrent testing of the method to cross reference the concentration testing with the two analytical techniques or even replicated measurements with the MC-ICPMS system. This could have (and should have) easily been done by sampling with two BAuT systems side-by-side to analyse the MC-ICPMS precision and the accuracy of the two methods. Passivation and memory effects of gold traps for Hg collection have been reported a lot in the literature and become an issue over long sampling durations (Brown et al., 2011; Gustin et al., 2011). While these sampling durations are shorter than other isotope collection methods, 3-24hrs is still a long time for gold trap sampling (5 mins for TGM concentration monitoring). It is likely that if there were some losses these would result in uptake in the breakthrough trap (hence the preference for using the same analysis method not to introduce bias into this analysis), but passivation also includes conversions and surface reactions that could potentially introduce some fractionation artefacts into the high precision isotope analyses. One method to control for this could be vaporising a known mass of SRM8610 into a known volume of Hg “zero” air and assessing the total Hg and isotope recoveries via this method?

Addressing some of these concerns (and specific comments below) may help the manuscript. Otherwise, I would recommend it for publication with minor corrections.

SPECIFIC COMMENTS:

Lines 36-38: the solution containing Hg2+ must first be reduced to Hg0

Lines 38-40: This is an incomplete sentence. Consider revising. Also the references used here should include recent publications by Jiskra et al. (2019) and Spzonar et al. (2020).

Line 41: “Having” should be “making”.

Lines 44-45: “…, but the method may not be suitable for a large amount of GEM.” why and according to who?

Lines 72: “RAR” is an unnecessary acronym and it is typically called inverse aqua regia. I would suggest just writing it out each time.

Lines 96-98: What does “covered by 2 CI and 3 confidence intervals” mean? Hg Isotopes should be summed by 2x S.D.

FIGURE 1: remove the outline around the whole figure. It doesn't look professional like this.

Section 2.2: The order of the methods is strange. Section 2.2 should really be before the analytical method (Section 2.1).

Line 116: “CE” is an unnecessary acronym. I would suggest just writing collection efficiency out each time.

Line 151: Please state the plastic polymer used in the Tedler bag.

TABLE 1: The tables are not very pleasant to look at and could be formatted much better - too many unnecessary lines. Also units in parenthesis. Applies to all tables.

TABLE 2: Really, there doesn't seem to be any problems with the recoveries of 6h and 12 h extractions. at 30-40% RAR. I'm not sure there is any advantage in the 24hr extraction method over these two.

Lines 226-247: This paragraph could really use some work. It is not easy to read and understand.

REFERENCES:

Brown, R. J. C., Kumar, Y., Brown, A. S., and Kim, K.-H.: Memory effects on adsorption tubes for mercury vapor measurement  in ambient air: elucidation, quantification, and strategies for mitigation of analytical bias, Environ. Sci. Technol., 45, 7812–7818, 2011.

Gustin, M. S., Lyman, S. N., Kilner, P., and Prestbo, E.: Development of a passive sampler for gaseous mercury, Atmos. Environ., 45, 5805–5812, 2011.

Jiskra, M., Marusczak, N., Leung, K. H., Hawkins, L., Prestbo, E., & Sonke, J. E.: Automated stable isotope sampling of gaseous elemental mercury (ISO-GEM): Insights into GEM emissions from building surfaces. Environmental science & technology, 53(8), 4346-4354, 2019.

Szponar, N., McLagan, D. S., Kaplan, R. J., Mitchell, C. P., Wania, F., Steffen, A., ... & Bergquist, B. A. Isotopic Characterization of Atmospheric Gaseous Elemental Mercury by Passive Air Sampling. Environmental Science & Technology, Accepted, doi.org/10.1021/acs.est.0c02251, 2020.

Reviewer 3 Report

This paper describes a fast sampling and extraction method for stable isotope measurement of gaseous mercury. The method developed in this research has a recovery of almost 100%. The analytical methods in the past often took days or even weeks. But there are some formatting problems, and the authors need to be corrected. 

1. Please correct the format of the table, do not copy then paste the EXCEL table format directly into the article.   2. How to ensure that the quartz materials used in the research (Figure 2) will not have mercury pollution problems?   3. Line 184, please provide the total power of the UV lamp.   4. Figure 3(b) is not related to the research topic, please delete.   5. Line 225, please rethink and correct the title of Section 3.3.   6. Please add more literature data in the Section 3 (Results and Discussion) and compare them with other analytical methods.  

Round 2

Reviewer 3 Report

n/a